# A common 1.6 mb Y-chromosomal inversion predisposes to subsequent deletions and severe spermatogenic failure in humans

Pille Hallast[1,2]*, Laura Kibena[1], Margus Punab[3,4], Elena Arciero[2], Siiri Rootsi[5], Marina Grigorova[1], Rodrigo Flores[5], Mark A Jobling[6], Olev Poolamets[3], Kristjan Pomm[3], Paul Korrovits[3], Kristiina Rull[1,4,7], Yali Xue[2], Chris Tyler-Smith[2†], Maris Laan[1†]*

[1]Institute of Biomedicine and Translational Medicine, University of Tartu, Tartu, Estonia; [2]Wellcome Genome Campus, Wellcome Sanger Institute, Hinxton, Cambridge, United Kingdom; [3]Andrology Unit, Tartu University Hospital, Tartu, Estonia; [4]Institute of Clinical Medicine, University of Tartu, Tartu, Estonia; [5]Institute of Genomics, Estonian Biocentre, University of Tartu, Tartu, Estonia; [6]Department of Genetics & Genome Biology, University of Leicester, Leicester, United Kingdom; [7]Women's Clinic, Tartu University Hospital, Tartu, Estonia

**Abstract** Male infertility is a prevalent condition, affecting 5–10% of men. So far, few genetic factors have been described as contributors to spermatogenic failure. Here, we report the first re-sequencing study of the Y-chromosomal *Azoospermia Factor c* (*AZFc*) region, combined with gene dosage analysis of the multicopy *DAZ*, *BPY2*, and *CDY* genes and Y-haplogroup determination. In analysing 2324 Estonian men, we uncovered a novel structural variant as a high-penetrance risk factor for male infertility. The Y lineage R1a1-M458, reported at >20% frequency in several European populations, carries a fixed ~1.6 Mb *r2/r3* inversion, destabilizing the *AZFc* region and predisposing to large recurrent microdeletions. Such complex rearrangements were significantly enriched among severe oligozoospermia cases. The carrier vs non-carrier risk for spermatogenic failure was increased 8.6-fold (p=$6.0\times10^{-4}$). This finding contributes to improved molecular diagnostics and clinical management of infertility. Carrier identification at young age will facilitate timely counselling and reproductive decision-making.

*For correspondence:
pille.hallast@ut.ee (PH);
maris.laan@ut.ee (ML)

†These authors contributed equally to this work

Competing interests: The authors declare that no competing interests exist.

## Introduction

The diagnosis of male factor infertility due to abnormal semen parameters concerns ~10% of men (*Jungwirth et al., 2012*; *Datta et al., 2016*). In today's andrology workup, ~60% of patients with spermatogenic failure remain idiopathic (*Punab et al., 2017*). Among the known causes, the most widely considered genetic factors are karyotype abnormalities (up to 17% of patients) and recurrent de novo microdeletions of the Y-chromosomal *Azoospermia Factor a* (*AZFa*) (~0.8 Mb), *AZFb* (~6.2 Mb), and *AZFc* (~3.5 Mb) regions (2–10%) (*Punab et al., 2017*; *Olesen et al., 2017*; *Tüttelmann et al., 2011*). For more than 15 years, testing for *AZF* deletions has been strongly recommended in the diagnostic workup for infertility patients with sperm concentration of <5 × $10^6$/ml (*ASRM, 2015*; *Krausz et al., 2014*). Most deletion carriers represent patients with either azoospermia (no sperm) or cryptozoospermia (>0–1 million sperm/ejaculate) (*Punab et al., 2017*; *Kohn et al., 2019*; *Stahl et al., 2010*). The most prevalent deletion type is *AZFc* (~80%), followed by the loss of *AZFa* (0.5–4%), *AZFb* (1–5%), and *AZFbc* (1–3%) regions (*Figure 1A*). Excess of recurrent *AZFc*

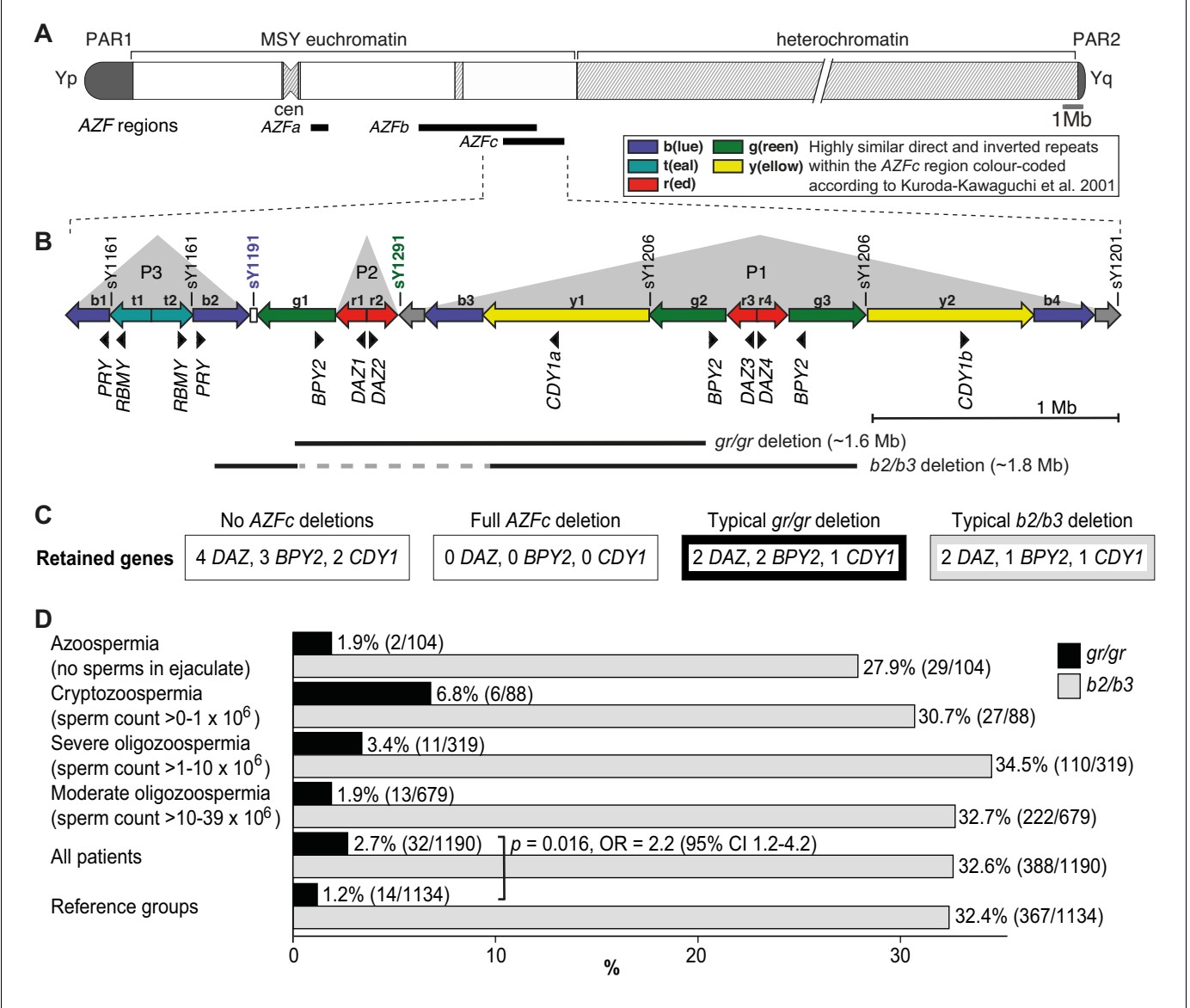

**Figure 1.** Y-chromosomal *AZFc* region and its partial deletions in the study group. (A) Schematic representation of the human Y chromosome with the *AZFa*, *AZFb*, and *AZFc* regions shown as black bars. (B) Magnified structure of the *AZFc* region with approximate locations of multicopy protein-coding genes, STS (sY) markers for the detection of *AZFc* partial deletions and the span of typical *gr/gr* and *b2/b3* deletions (*Kuroda-Kawaguchi et al., 2001*). P1–P3 (gray triangles) denote palindromic genomic segments consisting of two 'arms' representing highly similar inverted DNA repeats (>99.7% sequence identity) that flank a relatively short distinct 'spacer' sequence. Of note, the occurrence of the *b2/b3* deletion requires a preceding inversion in the *AZFc* region and therefore its presentation on the reference sequence includes also the retained segment (gray dashed line). Full details about alternative *gr/gr* and *b2/b3* deletion types are presented in *Figure 1—figure supplement 1*. (C) Dosage of multicopy genes on human Y chromosomes with or without *AZFc* deletions. (D) Prevalence of the *gr/gr* and *b2/b3* deletions detected in the subgroups of this study. Fisher's exact test was used to test the statistical significance in the deletion frequencies between the groups. PAR, pseudoautosomal region; MSY, male-specific region of the Y chromosome; cen, centromere; *AZF*, azoospermia factor region.

The online version of this article includes the following figure supplement(s) for figure 1:

**Figure supplement 1.** The human Y chromosome and the *Azoospermia Factor (AZF)* regions.

deletions is promoted by the region's complex genomic structure comprised of long direct and inverted amplicons of nearly identical DNA segments that lead to aberrant meiotic rearrangements in gametogenesis (*Kuroda-Kawaguchi et al., 2001*; *Skaletsky et al., 2003*; *Figure 1B*). The *AZFc* full deletions remove all the multicopy *DAZ (deleted in azoospermia 1)*, *BPY2 (basic charge Y-linked*

2), and *CDY1* (*chromodomain Y-linked 1*) genes that are expressed in a testis-enriched manner and considered important in spermatogenesis (*Figure 1C*).

The palindromic structure of the *AZFc* region also facilitates partial deletions that are rather frequently detected in the general population (*Repping et al., 2003*; *Repping et al., 2004*; *Rozen et al., 2012*; *Fernandes et al., 2004*). The most prevalent partial deletion types, named after the involved amplicons as *g(reen)-r(ed)/g(reen)-r(ed)* (lost segment ~1.6 Mb) and *b(lue)2/b(lue)3* (~1.8 Mb) reduce the copy number of *DAZ*, *BPY2*, and *CDY1* genes by roughly 50% (*Figure 1B,C*, *Figure 1—figure supplement 1*). The published data on the contribution of *gr/gr* and *b2/b3* deletions to spermatogenic failure are inconsistent. In European populations, the carrier status of the *gr/gr* deletion increases a risk to low sperm counts ~1.8-fold (*Rozen et al., 2012*; *Bansal et al., 2016a*; *Krausz et al., 2009*). Its more variable effect on spermatogenesis has been shown in Middle Eastern and Asian populations, where the *gr/gr* deletion is completely fixed in some Y lineages, for example haplogroups D2 and Q1a that are common in Japan and some parts of China (*de Carvalho et al., 2006*; *Teitz et al., 2018*). In contrast, the *b2/b3* deletion appears to be a risk factor for spermatogenic impairment in several East Asian and African, but not in European or South Asian populations (*Bansal et al., 2016b*; *Colaco and Modi, 2018*). Notably, the *b2/b3* deletion is completely fixed in Y haplogroup N3 that has a high frequency (up to 90% in some populations) in Finno-Ugric-, Baltic-, and some Turkic-speaking people living in Northern Eurasia (*Rozen et al., 2012*; *Fernandes et al., 2004*; *Ilumäe et al., 2016*). Thus, it is unlikely that the carriership of a *gr/gr* or *b2/b3* deletion per se has an effect on male fertility potential. It has been proposed that this broad phenotypic variability may be explained by the diversity of *gr/gr* and *b2/b3* deletion subtypes (*Machev, 2004*). Y chromosomes carrying partial *AZFc* deletions may differ for the content, dosage, or genetic variability of the retained genes, the overall genetic composition reflected by phylogenetic haplogroups or the presence of additional structural variants. Only limited studies have analyzed the subtypes of *gr/gr* or *b2/b3* deletions, and no straightforward conclusions have been reached for their link to spermatogenic failure (*Krausz et al., 2009*; *Ghorbel et al., 2016*; *Krausz and Casamonti, 2017*).

The current study represents the largest in-depth investigation of *AZFc* partial deletions in men recruited by a single European clinical center. We analysed 1190 Estonian idiopathic patients with male factor infertility in comparison to 1134 reference men from the same population, including 2000 subjects with sperm parameter data available. Y chromosomes carrying *gr/gr* or *b2/b3* deletions were investigated for additional genomic rearrangements, Y-chromosomal haplogroups, and dosage and sequence variation of the retained *DAZ*, *BPY2*, and *CDY* genes. The study aimed to determine the role and contribution of *gr/gr* and *b2/b3* deletion subtypes in spermatogenic failure and to explore their potential in the clinical perspective.

## Results

### Enrichment of *gr/gr* deletions in Estonian idiopathic infertile men with reduced sperm counts

The study analyzed 1190 Estonian men with idiopathic infertility (sperm counts $0–39 \times 10^6$/ejaculate) and a reference group comprised of 1134 Estonian men with proven fatherhood (n = 635) or representing healthy young men (n = 499) (*Table 1*, *Supplementary file 1*). For all 2324 study subjects, complete *AZFa*, *AZFb*, and *AZFc* deletions were excluded.

The partial *AZFc* deletions identified using the STS-based polymerase chain reaction (PCR) assays were *gr/gr* (n = 46), *b2/b3* (n = 756), and *b1/b3* (n = 1, reference case) (*Table 2*). A statistically significant excess of *gr/gr* deletions was detected in idiopathic male infertility patients (2.7%; n = 32/1190) compared to reference cases (1.2%; n = 14/1134) (Fisher's exact test, p=0.016; odds ratio [OR] = 2.2 [95% confidence interval (CI) 1.2–4.2]) (*Figure 1D*, *Supplementary file 2*). The highest frequency of *gr/gr* deletion carriers (6.8%, n = 6/88) was detected in cryptozoospermia cases (sperm count $> 0–1 \times 10^6$/ejaculate). However, in the reference group andrological parameters of men with or without the *gr/gr* deletion did not differ (*Supplementary file 3*). All 10 reference men with the *gr/gr* deletion and available andrological data were normozoospermic (220.3 [74.2–559.0] $\times 10^6$ sperm/ejaculate). Also their other andrological parameters were within the normal range, overlapping with those of the subjects without a *gr/gr* deletion.

**Table 1.** Characteristics of the patients with male factor infertility and reference groups used for comparison.

| Parameter | Unit | Idiopathic spermatogenic impairment (n = 1190)* | | | Reference groups (n = 1134) | | |
| | | Azoo-/cryptozoospermia | Severe oligozoospermia | Moderate oligozoospermia | Partners of pregnant women[†] | Estonian young men cohort[‡] | REPROMETA proven fathers[§] |
|---|---|---|---|---|---|---|---|
| | n | 104/88 | 319 | 679 | 324 | 499 | 311 |
| Age | Years | 33.2 (23.6–51.8) | 32.2 (23.9–49.5) | 31.7 (23.0–44.6) | 31.0 (22.9–45.0) | 18.6 (17.2–22.9) | 31.0 (21.0–43.0) |
| BMI | $kg/m^2$ | 26.0 (21.2–34.4) | 25.9 (20.2–35.5) | 25.8 (20.1–34.6) | 24.8 (20.0–32.2) | 22.0 (18.7–27.5) | 25.9 (20.2–33.1) |
| Total testis volume | ml | 33.5 (17.0–49.0) | 39.0 (22.0–50.0) | 40.0 (26.0–52.0) | 46.0 (34.0–62.4) | 50.0 (35.0–70.0) | n.d. |
| Semen volume | ml | 3.3 (0.8–6.6) | 3.3 (1.1–7.0) | 3.6 (1.6–6.9) | 3.7 (1.7–8.0) | 3.2 (1.2–6.4) | n.d. |
| Sperm concentration | $\times 10^6$/ml | 0 (0–0.2) | 1.4 (0.4–5.2) | 6.0 (2.2–15.2) | 76.0 (16.7–236.0) | 66.8 (8.2–225.1) | n.d. |
| Total sperm count | $\times 10^6$/ ejaculate | 0 (0–0.7) | 4.7 (1.3–9.3) | 23.1 (11.0–37.5) | 295.2 (60.0–980.1) | 221.6 (18.4–788.0) | n.d. |
| Progressive A+B motility | % | 0 (0–37.2) | 16.0 (0–47.2) | 27.0 (1.0–57.0) | 50.0 (30.0–69.0) | 57.3 (34.7–75.3) | n.d. |
| Sperms with normal morphology | % | 0 (0–1.0) | 0 (0–6.0) | 2.0 (0–9.0) | 10.0 (2.0–19.1) | 12.0 (4.0–20.0) | n.d. |
| FSH | IU/l | 13.7 (2.7–38.2) | 6.6 (1.9–22.8) | 5.2 (1.8–16.5) | 3.6 (1.5–8.3) | 2.8 (1.2–6.7) | n.d. |
| LH | IU/l | 5.7 (2.1–12.0) | 4.6 (1.9–9.9) | 4.2 (1.8–8.4) | 3.6 (1.5–6.7) | 3.8 (1.8–7.2) | n.d. |
| Total testosterone | nmol/l | 15.3 (7.7–28.4) | 16.6 (7.9–30.0) | 16.6 (8.5–30.3) | 16.5 (8.8–27.2) | 27.7 (15.4–46.3) | n.d. |

All study subjects were recruited in Estonia. For each parameter, median and (5th–95th) percentile values are shown. Additional details in *Supplementary file 1*.

*Patients were subgrouped based on total sperm counts per ejaculate: azoospermia, no sperm; cryptozoospermia, sperm counts > 0–1 × 10^6; severe oligozoospermia, >1–10 × 10^6; moderate oligozoospermia, >10–39 × 10^6 (**Punab et al., 2017**).

[†]Male partners of pregnant women (**Punab et al., 2017**); eight men had sperm counts < 39 × 10^6; for four men, sperm analysis was not available.

[‡]Male cohort without fatherhood data (**Grigorova et al., 2008**); 47 men had sperm counts < 39 × 10^6; for nine men, sperm analysis was not available.

[§]REPROMETA study recruited and sampled couples after delivery of their newborn; details in **Kikas et al., 2020**; **Pilvar et al., 2019**.

n.d., not determined.

The patient and the reference groups exhibited similar prevalence of *b2/b3* deletions (388/1190, 32.6% vs 367/1134, 32.4%; Fisher's exact test, p=0.8). No apparent clinically meaningful genetic effects on andrological parameters were observed in either of the study groups (*Supplementary files 3* and *4*).

## Significant overrepresentation of Y lineage R1a1-M458 in *gr/gr* deletion carriers

The Y-chromosomal haplogroups determined by typing phylogenetically informative markers in 31 patients and 13 reference men carrying a *gr/gr* deletion represented 20 different lineages (patients, 17; reference men, 10; *Figure 2A*, *Supplementary file 5*). Combining the phylogenetic context with the data on exact missing *DAZ* and *CDY1* gene copies (see below) revealed that the *gr/gr* deletion events in 44 analyzed cases must have independently occurred at least 26 times. About two-thirds of these Y chromosomes belonged to haplogroup R1, whereas the rest represented A1b, G, I, and J lineages. Notably, there was a highly significant overrepresentation of Y chromosomes belonging to lineage R1a1-M458 in the *gr/gr* deletion carriers compared to the known Estonian population frequency (22.7% vs 5.1%; Fisher's exact test, p=5.3×10^{−4}, OR = 5.5 [95% CI 2.2–13.7]; *Figure 2B*, *Supplementary file 5*; *Underhill et al., 2015*).

Nearly all (99.4%) Estonian cases with the *b2/b3* deletion belonged to the Y haplogroup N3, in which this event is fixed (*Repping et al., 2004*; *Fernandes et al., 2004*). The most commonly

**Table 2.** Summary of the identified Y-chromosomal *AZF* deletion subtypes.

| Y-chromosomal rearrangements | Idiopathic male infertility patients (n) | Reference men (n) |
|---|---|---|
| All analyzed cases | 1190 | 1134 |
| Any *AZFc gr/gr* deletion | 32 (2.7%) | 14 (1.2%) |
| | Fisher's exact test, p=0.016; OR = 2.2 [95% CI 1.2–4.2] | |
| Any *AZFc b2/b3* deletion | 388 (32.6%) | 367 (32.4%) |
| Other type of *AZF* deletion | Loss of *b2/b3* marker sY1191 (one case) | *AZFc b1/b3* del (one case); partial *AZFa* del (one case) |
| No deletion | 769 (64.6%) | 751 (66.2%) |
| *Simple partial AZFc deletions* | | |
| Typical *gr/gr* deletion | 19/31 (61.3%) | 8/13 (61.5%) |
| Typical *b2/b3* deletion* | 300/382 (78.5%) | 210/249 (84.3%) |
| *AZFc partial deletion followed by b2/b4 duplication* | | |
| *gr/gr* del + *b2/b4* dupl[†] | 2/31 (6.5%) | 3/13 (23.1%) |
| | Fisher's exact test, p=0.144; OR = 0.2 [95% CI 0.0–1.6] | |
| *b2/b3* del + *b2/b4* dupl[*,†] | 78/382 (20.4%) | 34/249 (13.7%) |
| | Fisher's exact test, p=0.026; OR = 1.6 [95% CI 1.0–2.4] | |
| *AZFc partial deletion and atypical genomic rearrangements*[‡] | | |
| *gr/gr* del + extra gene copies | 1 | 0 |
| *b2/b3* del + extra gene copies | 3 | 4 |
| *Complex events on the Y lineage R1a1-M458 with the preceding AZFc r2/r3 inversion* | | |
| *r2/r3* inv + *gr/gr* del | 8 | 2[§] |
| *r2/r3* inv + *gr/gr* del + *b2/b4* dupl | 1 | 0 |
| *r2/r3* inv + loss of marker sY1191 + secondary gene duplications[¶] | 1 | 0 |
| *r2/r3* inv + *b2/b3* del + *b2/b4* dupl | 0 | 1** |
| *Carriers of any AZFc gr/gr deletion type without the preceding r2/r3 inversion* | | |
| *gr/gr* del w/o detected *r2/r3* inv | 23/1190 (1.9 %) | 12/1134 (1.1%) |
| | Fisher's exact test, p=0.090; OR = 1.8 [95% CI 0.9–3.7] | |

*Deletion subtype analysis was carried out for cases with available sufficient quantities of DNA. REPROMETA subjects were excluded from the *b2/b3* deletion subtype analysis and subsequent statistical testing due to missing andrological data.

[†]One or more amplicons of the retained '2x*DAZ*, 2x*BPY2*, 1x*CDY1*' (*gr/gr* deletion) or '2x*DAZ*, 1x*BPY2*, 1x*CDY1*' (*b2/b3* deletion) genes.

[‡]Additional copies of *DAZ*, *BPY2*, and/or *CDY1* genes inconsistent with the full '*b2/b4*' duplication.

[§]Including one REPROMETA man without andrological data.

[¶]Detected gene copy numbers 6x*DAZ*, 4x*BPY2*, 3x*CDY1*; the obligate presence of *r2/r3* inversion was defined based on Y-chromosomal phylogeny as the man carries Y lineage R1a1a1b1a1a1c-CTS11962.1 that was also identified in two cases with the *r2/r3* inversion (**Supplementary file 12**).

**Man from 'Partners of pregnant women' cohort with sperm concentration $12 \times 10^6$/ml below normozoospermia threshold ($15 \times 10^6$/ml) and sperm counts $39.4 \times 10^6$/ejaculate at the borderline of the lowest reference value ($39.0 \times 10^6$/ejaculate).

detected sub-lineage was N3a3a-L550 (~51% of 436 typed chromosomes) and in total 15 different haplogroups that had diverged after the *b2/b3* deletion event in the common ancestor of N3 were present in Estonian men (**Figure 2C**, **Supplementary file 6**). *b2/b3* Y chromosomes representing non-N3 lineages were detected in two patients and three reference men. Lineage typing was possible for three of them, who carried either K-M9 (one patient) or R1a1a1b1a1a1c-CTS11962.1 (one patient and one reference case).

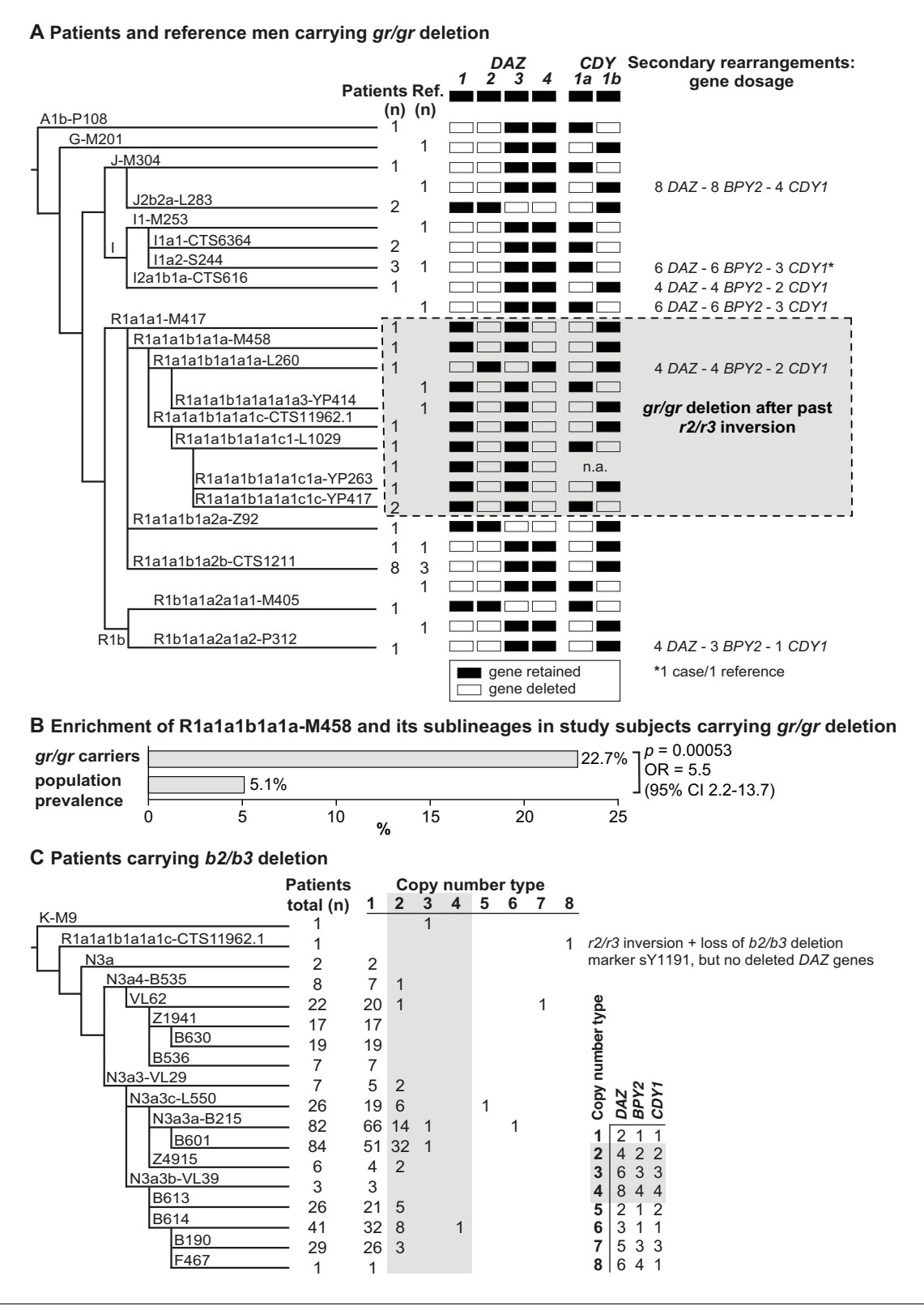

**Figure 2.** Phylogenetic relationships and gene copies in study subjects with partial *AZFc* deletions. (**A**) Y-chromosomal lineages indicated with typed terminal markers (left), deleted (white)/retained (black) *DAZ* and *CDY1* gene copies (middle), and secondary rearrangements in the *AZFc* region (right) of idiopathic male factor infertility (n = 31) and reference cases (n = 13) carrying the *gr/gr* deletion. The human Y-chromosomal reference sequence has four *DAZ* and two *CDY1* copies; the retained gene copies on each Y chromosome with a *gr/gr* deletion are shown as filled boxes. Chromosomes
*Figure 2 continued on next page*

**eLife** Research article

Evolutionary Biology | Genetics and Genomics

*Figure 2 continued*

carrying atypical *gr/gr* subtypes with the loss of either the *DAZ1/DAZ3* or *DAZ2/DAZ4* gene pair due to complex genomic rearrangement combining the previous *r2/r3* inversion with a subsequent *gr/gr* deletion are highlighted with a dashed gray square. (B) Enrichment of the Y-chromosomal lineage R1a1-M458 and its sub-lineages in study subjects carrying the *gr/gr* deletion in comparison to the Estonian general population (data from *Underhill et al., 2015*). Fisher's exact test was used to test the statistical significance between the groups. (C) Y-chromosomal lineages indicated with typed terminal markers (left) and the copy number of the *DAZ*, *BPY2*, and *CDY1* gene copies (right) determined for 382 idiopathic male factor infertility cases carrying the *b2/b3* deletion. The light gray box denotes *DAZ*, *BPY2*, and *CDY1* gene dosage consistent with full *b2/b4* duplication(s). The legend for the deletion subtype is shown in the bottom right corner. Further information on the distribution of Y-chromosomal lineages in the carriers of *AZFc* partial deletions are provided in *Supplementary files 5* and *6*, and the *AZFc* rearrangement types are detailed in *Figure 1—figure supplement 1* and *Supplementary files 7* and *9–12*. n, number; n.a., not available; Ref, reference cases.

The online version of this article includes the following figure supplement(s) for figure 2:

**Figure supplement 1.** Histograms of estimated raw copy number values for *DAZ*, *BPY2*, and *CDY* genes by ddPCR.

**Figure supplement 2.** Location of the identified exonic variants in the *DAZ* genes.

## Increased prevalence of *b2/b3* deletion followed by *b2/b4* duplication in infertile men

The expected retained copy number of *DAZ*, *BPY2*, and *CDY1* genes consistent with the typical *gr/gr* deletion, as determined by quantification using Droplet Digital PCR (ddPCR), was found in 37/44 (~84%) cases (*Figure 2A*, *Figure 2—figure supplement 1*, *Table 2*, *Supplementary file 7*). Three patients and three reference men carried a secondary *b2/b4* duplication adding one or more amplicons of [two *DAZ* – two *BPY2* – one *CDY1*] genes with no apparent effect on infertility status (Fisher's exact test, p=0.34). Notably, four of six samples with secondary *b2/b4* duplication events were identified in haplogroup I. This complex rearrangement has also been reported in the gnomAD SV database (v 2.1) in 114/5528 analyzed men from around the world with the prevalence of 3.5% in East Asians and 1.2% in Europeans (*Supplementary file 8*; *Collins et al., 2020*).

Similarly, 78.5% of patients and 84.3% reference men with the *b2/b3* deletion presented gene dosage consistent with the typical deletion (*Figure 2C*, *Figure 2—figure supplement 1*, *Table 2*, *Supplementary files 7 and 9*). Indicative of recurrent secondary events, one or more *b2/b4* duplications of [two *DAZ* – one *BPY2* – one *CDY1*] genes were identified in 13 haplogroups, including non-N3 lineages K-M9 and R1a1a1b1a1a1c-CTS11962.1. In the gnomAD, 58/5115 men have been reported with this duplication, with the prevalence of 2.2% in East Asians and 0.9% in Europeans (*Supplementary file 8*). Although secondary *b2/b4* duplications were detected with significantly higher prevalence in patients compared to the reference men (n = 78/382, 20.4% vs n = 35/249, 14.1%; Fisher's exact test, p=0.026, OR = 1.57 [95% CI 1.02–2.42]), no consistent effect of increased gene copy number on andrological parameters was observed (*Supplementary files 3* and *4*). Reference men with *b2/b4* duplication compared to subjects with no *AZFc* rearrangements showed a trend for lower follicle-stimulating hormone (FSH) (median 2.3 [5–95% range 1.4–7.5] vs 3.2 [1.3–7.1] IU/l; p<0.05) and luteinizing hormone (LH) (3.1 [1.7–5.0] vs 3.8 (1.7–7.2) IU/l; p<0.05). Additionally, in eight subjects with *AZFc* partial deletions, further atypical Y-chromosomal genomic rearrangements were detected, but also with no clear evidence for a phenotypic effect (*Table 2*, *Supplementary file 7*).

The data gathered from this analysis thus suggest that the dosage of *DAZ*, *BPY2*, and *CDY1* genes does not play a major role in modulating the pathogenic effect of the *gr/gr* and *b2/b3* deletions.

## No specific *DAZ* or *CDY1* gene copy is lost in men with spermatogenic failure

The deletion subtypes for *b2/b3* and *gr/gr* carriers were identified by determining the genotypes of *DAZ* and *CDY1* gene-specific paralogous sequence variants. The major *b2/b3* deletion subtype in both patients (99.7%) and reference cases (98.1%) was the loss of *DAZ3-DAZ4-CDY1a* genes, whereas the most frequent *gr/gr* subtypes among all the deletion carriers were the loss of *DAZ1-DAZ2-CDY1a* (41.9%, 18/43 cases) and *DAZ1-DAZ2-CDY1b* (25.6%, 11/43 cases) combinations (*Figure 2A*, *Supplementary files 10* and *11*). The observed prevalence of the major *gr/gr* subtypes was concordant with the published data on other European populations (42.5% and 25.5%, respectively; *Krausz et al., 2009*). As these *gr/gr* deletion subtypes are prevalent in the reference group

(total 11 of 13, 84.6%), their major role in spermatogenic impairment can be ruled out. As a novel insight, a subset of these Y chromosomes showed lineage-specific loss of some exon 7 subtypes of the retained *DAZ4* gene (*Figure 2—figure supplement 2*; *Supplementary file 11*). All five exons 7Y in the *DAZ4* gene were missing in the Y chromosomes with the *DAZ1-DAZ2-CDY1a* deletion that had occurred in sub-lineages of the R1a1a1b1a2 haplogroup (9/31 patients, 3/13 reference men), representing 12/18 *DAZ1-DAZ2-CDY1a* deletion carriers. The exon 7F in *DAZ4* was lost in haplogroup I1 and its sub-lineages (5/31, 2/13), that is 7/11 individuals carrying the *DAZ1-DAZ2-CDY1b* deletion (*Supplementary files 10* and *11*). There was no evidence that loss of *DAZ4* exons 7Y or 7F has any phenotypic consequences. Most likely, this observation reflects gene conversion events from *DAZ3* to *DAZ4* as the former lacks both, exons 7Y and 7F.

Taken together, our findings indicate that neither the loss of the *DAZ1-DAZ2* nor the *DAZ3-DAZ4* gene pair, combined with either a *CDY1a* or *CDY1b* gene, directly causes spermatogenic failure. Interestingly, no Y chromosomes were observed with fewer than two retained *DAZ* genes.

## Y lineage R1a1-M458 carries a fixed *r2/r3* inversion predisposing to recurrent deletions

Novel atypical *gr/gr* and *b2/b3* deletion subtypes with the loss of an unusual *DAZ* gene pair were identified (*Figure 2A*, *Table 2*, *Supplementary files 10–12*). Eight patients and two reference cases with a *gr/gr* deletion were missing *DAZ2-DAZ4* genes. Loss of *DAZ1-DAZ3* genes followed by a subsequent *b2/b4* duplication event was identified in one infertile and one reference case with either *gr/gr* or *b2/b3* deletion, respectively. All but one subject with this atypical pair of lost *DAZ* genes belonged to the Y haplogroup R1a1-M458 and its sub-lineages, significantly enriched in *gr/gr* deletion carriers (*Figure 2B*, *Supplementary file 5*). The most parsimonious explanation to explain the simultaneous deletion of either *DAZ1-DAZ3* or *DAZ2-DAZ4* genes is a preceding ~1.6 Mb long inversion between the *r(ed)two* and *r(ed)three* amplicons (*Figure 3A*). This new inverted structure might be more susceptible to recurrent deletions as it has altered the internal palindromic structure of *AZFc* region. In *r2/r3* inversion chromosomes, the largest palindrome P1 is almost completely lost and the size of the palindrome P2 is greatly expanded by positioning the homologous *g1/g2* segments in an inverted orientation. The *r2/r3* inversion is consequently expected to destabilize the *AZFc* region as several long DNA amplicons with highly homologous DNA sequence are positioned in the same sequence orientation (*b2*, *b3*, and *b4*; *g2* and *g3*; *y1* and *y3*). Therefore, they are prone to non-allelic homologous recombination mediating recurrent deletions and duplications. Since these atypical deletion subtypes were identified only in a specific Y-chromosomal haplogroup, the detected *r2/r3* inversion must have occurred only once in the common ancestor of R1a1-M458 sublineages. One patient with the loss of *DAZ2-DAZ4* carried haplogroup R1a1a1-M417, an ancestral lineage to R1a1-M458 (*Figure 2A*). However, lineage R1a1a1-M417 is not fixed for this inversion since its other sub-lineage, R1a1a1b1a2, does not carry it and any subsequent inversion restoring the exact original *AZFc* structure is not credible. The more parsimonious explanation is that the inversion occurred in a sub-lineage of R1a1a1-M417 that has to be yet determined.

Based on the Y-chromosomal phylogenetic data, one additional patient was identified as an obligate carrier of the *r2/r3* inversion as his Y chromosome represents the lineage R1a1a1b1a1a1c-CTS11962.1 that was also identified in two cases with the *r2/r3* inversion. This patient exhibited signs of unusual deletion and duplication events in the *AZFc* region as he carried six *DAZ*, four *BPY2*, and three copies of the *CDY1* gene (*Figure 2B*, *Table 2*, *Supplementary file 9*).

Among the analyzed 2324 men, 13 cases with the complex *AZFc* rearrangement combining *r2/r3* inversion with a subsequent deletion (from here on referred to as '*r2/r3* inversion plus deletion' for simplicity), represented 0.6% (*Table 3*). Considering the reported population prevalence of R1a1-M458 lineage in Estonians (5.1%; *Underhill et al., 2015*), the estimated number of subjects representing this Y lineage in the study group was ~119. Thus, approximately one in ten chromosomes with the *r2/r3* inversion had undergone a subsequent deletion event (13/119, 11%).

## *r2/r3* inversion promotes recurrent deletions that lead to severe oligoasthenoteratozoospermia

Idiopathic infertility cases carrying the *r2/r3* inversion plus deletion in the *AZFc* region (n = 10) exhibited extremely low sperm counts compared to subjects without any *AZFc* deletions (median 2.0 vs

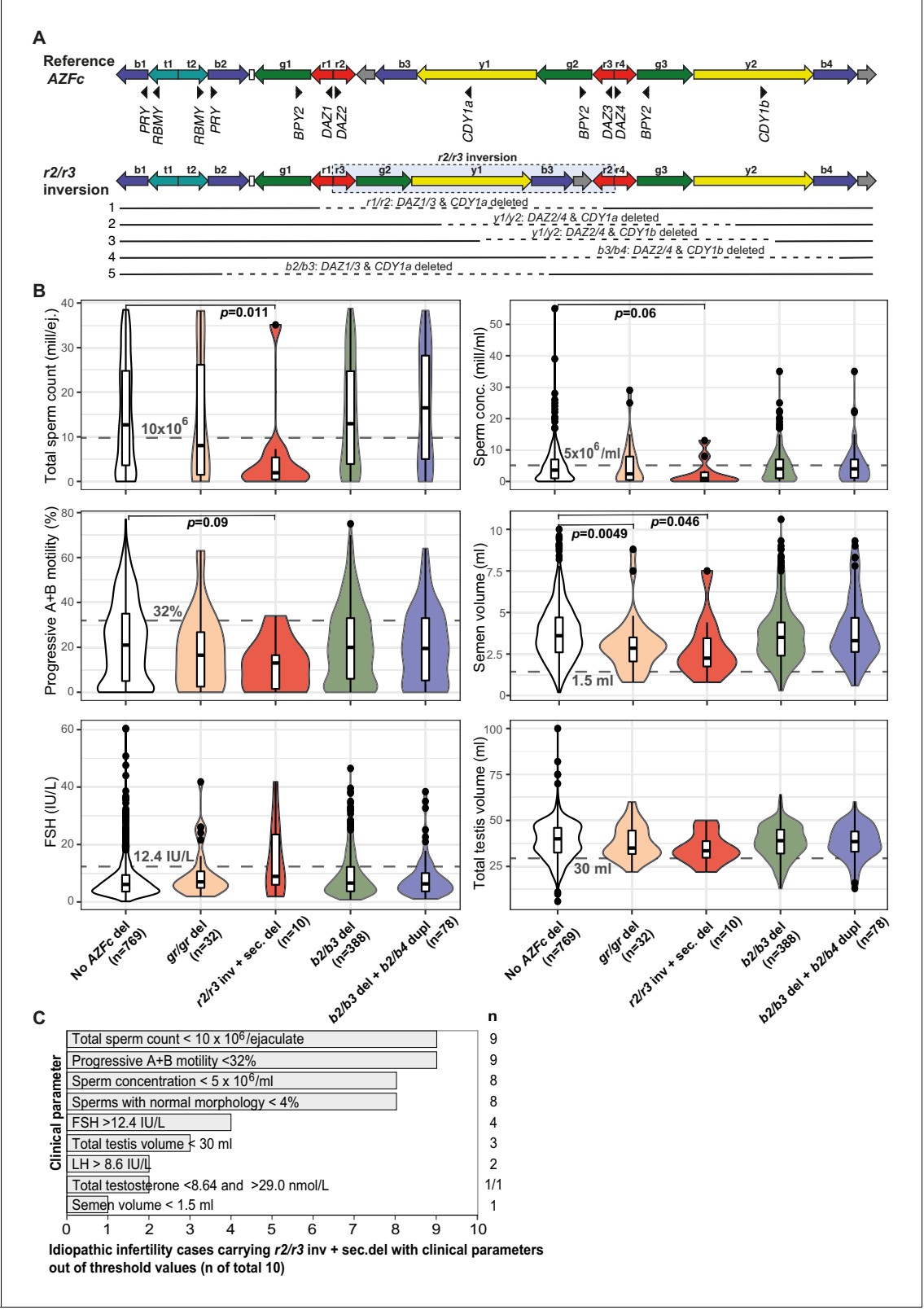

**Figure 3.** Complex structural variants at the Y-chromosomal lineage R1a1-M458 and their effect on andrological parameters. (**A**) Schematic presentation of the Y chromosome with the *r2/r3* inversion compared to the reference sequence. The *r2/r3* inversion structure nearly destroys the large palindrome P1 and, consequently, destabilizes the *AZFc* region since several long DNA amplicons with highly similar DNA sequence (*b2, b3*, and *b4*; *g2* and *g3*; *y1* and *y3*) are positioned in the same sequence orientation. This structure promotes non-allelic homologous recombination mediating

*Figure 3 continued on next page*

*Figure 3 continued*

recurrent deletion and duplication events. The approximate regions removed by the identified *gr/gr* and *b2/b3* deletions arising on the *r2/r3* inverted Y chromosome are shown as dashed lines. (B) Distribution of andrological parameters in the idiopathic male factor infertility cases (total sperm counts 0–39 × $10^6$) subgrouped based on the structure of the *AZFc* region. The pairwise Wilcoxon rank-sum test was applied to estimate the statistical difference between groups (Bonferroni threshold for multiple testing correction, p<1.0×$10^{-3}$). Threshold values (shown in gray) for sperm parameters corresponding to severe spermatogenic failure are based on international guidelines (*World Health Organization, 2010*). For reproductive hormones, reference values of the laboratory service provider are shown. The empirical threshold for the total testis volume was based on routinely applied clinical criteria at the AC-TUH. For additional details, see *Figure 3—figure supplements 1–3*, *Supplementary files 3* and *12*. (C) The majority of idiopathic infertility cases carrying the *r2/r3* inversion plus secondary *AZFc* partial deletions (total n = 10) exhibit severe oligoasthenoteratozoospermia (OAT) defined as extremely reduced sperm counts (<5 × $10^6$/ml) and concentration (<10 × $10^6$/ejaculate) combined with low fraction of sperms with normal morphology (<4% normal forms) and motility (<32% progressive motile spermatozoa). Reference values for andrological parameters have been applied as referred in (B). As total testis volume is mostly within the expected range, their infertility is not caused by intrinsic congenital testicular damage but rather due to severe spermatogenic failure per se. Del, deletion; inv, inversion; dupl, duplication; n, number; sec, secondary; mill, million; ej., ejaculate.

The online version of this article includes the following figure supplement(s) for figure 3:

**Figure supplement 1.** Distribution of seminal parameters in idiopathic male factor infertility cases with spermatogenic impairment and reference subjects.

**Figure supplement 2.** Distribution of hormonal and testicular parameters in idiopathic male factor infertility cases with spermatogenic impairment and reference individuals.

**Figure supplement 3.** Distribution of andrological parameters in the idiopathic male factor infertility cases (total sperm counts 0–39 × $10^6$) subgrouped based on the structure of the *AZFc* region.

12.5 × $10^6$/ejaculate; Wilcoxon test, nominal p=0.011) (*Figure 3B*, *Figure 3—figure supplements 1–3*, *Supplementary file 3*). Nine of 10 men showed severe spermatogenic failure (total sperm counts <10 × $10^6$/ejaculate), either azoospermia (n = 1), cryptozoospermia (n = 3), or severe oligozoospermia (n = 5) (*Supplementary files 11* and *12*). They also showed consistently the poorest sperm concentration (median 1.0 × $10^6$/ml) and progressive motility (13%), as well as the lowest semen volume (2.3 ml) compared to the rest of analyzed infertile men. The data suggests that extreme oligoasthenoteratozoospermia (OAT) observed in these subjects was due to the severely affected process of spermatogenesis, whereas their testicular volume and hormonal profile were within the typical range of male factor infertility cases (*Figure 3C*).

**Table 3.** Enrichment of the *AZFc r2/r3* inversion followed by a partial *AZFc* deletion in men with severe spermatogenic failure.

| Group | All (n) | *AZFc r2/r3* inversion + *AZFc* partial deletion | | |
| | | Estimated non-carriers (n) | Detected carriers (n) | % of carriers in the (sub)group |
|---|---|---|---|---|
| *a. Full study group* | | | | |
| All analyzed study subjects | 2324 | 2311 | 13 | 0.6% |
| Study subjects with sperm counts | 2000 | 1988 | 12 | 0.6% |
| *Subjects stratified based on total sperm counts per ejaculate* | | | | |
| Sperm counts 0–10 × $10^6$ | 524 | 515 | 9 | 1.7% |
| Sperm counts > 10 × $10^6$ | 1476 | 1473 | 3 | 0.2% |
| | | Fisher's exact test, p=6.0×$10^{-4}$, OR = 8.6 [95% CI 2.3–31.8] | | |
| *b. Carriers of the Y lineage R1a1a-M458** | | | | |
| In all analyzed study subjects | 119 | 106 | 13 | 11.0% |
| In study subjects with sperm counts | 102 | 90 | 12 | 11.8% |
| *Subjects stratified based on total sperm counts per ejaculate* | | | | |
| Sperm counts 0–10 × $10^6$ | 27 | 18 | 9 | 33.7% |
| Sperm counts > 10 × $10^6$ | 75 | 72 | 3 | 4.0% |
| | | Fisher's exact test, p=3.0×$10^{-4}$, OR = 12.0 [95% CI 2.9–48.9] | | |

*Expected number of Y lineage R1a1-M458 in each subgroup was estimated using the known Estonian population prevalence 5.1% (*Underhill et al., 2015*).

When all the men with andrological data (n = 2000) were stratified based on sperm counts, there was a highly significant enrichment of *r2/r3* inversion plus deletion in men with severe spermatogenic failure (sperm counts $0–10 \times 10^6$) compared to the rest (1.7% vs 0.2%, Fisher's exact test, p=$6.0\times10^{-4}$, OR = 8.6 [95% CI 2.3–31.8]; *Table 3*). The estimated number of phenotyped subjects representing the Y haplogroup R1a1-M458 with the fixed *r2/r3* inversion was 102 (based on population prevalence 5.1%; *Underhill et al., 2015*). Among carriers of this Y lineage, 33.7% of men with sperm counts $0–10 \times 10^6$ (9/27), but only 4.0% with sperm counts of $>10 \times 10^6$ (3/72) had undergone a subsequent *AZFc* partial deletion (Fisher's exact test, p=$3.0\times10^{-4}$, OR = 12.0 [95% CI 2.9–48.9]).

Only three reference cases carried a Y chromosome with the *r2/r3* inversion plus deletion. At the time of phenotyping, all three subjects were younger (aged 18, 21, and 23 years) than the variant carriers in the idiopathic infertility group (median 32.4, range 26–51 years) (*Supplementary file 12*). The only reference subject with this complex *AZFc* rearrangement, but unaffected sperm analysis was the youngest (18 years). Notably, another reference man (23 years) with andrological data would actually be classified, based on WHO guidelines (*World Health Organization, 2010*), as an oligozoospermia case (sperm concentration $12 \times 10^6$/ml vs threshold $15 \times 10^6$/ml). Also, his total sperm counts ($39.4 \times 10^6$/ejaculate) represented a borderline value.

## Sequence diversity of the retained *DAZ*, *BPY2*, and *CDY* genes is extremely low and has no detectable effect on sperm parameters

The re-sequenced retained *DAZ1-4*, *BPY2*, and *CDY1-2* genes were characterized by extremely low nucleotide variability in all Y-chromosomal lineages and deletion subtypes (*Supplementary file 13*). For 476 samples (*gr/gr*, n = 40; *b2/b3*, n = 436), re-sequenced for the >94 kb region using Illumina MiSeq, a total of 42 variants were identified with median 0.8 variants/kb and maximum two variants per individual. Most of them were previously undescribed (*Giachini et al., 2008*), singletons (*Jobling and Tyler-Smith, 2017*), and/or non-coding SNVs/short indels (*Lu et al., 2011*; *Supplementary file 14*). The *CDY2a-CDY2b* genes harbored only one variable site, whereas *DAZ1-DAZ2* carried 24 or 26 SNVs/indels. Most variants appeared paralogous as both the reference and alternative alleles were identified. Among the four detected missense variants, CDY1b p.T419N was fixed in all three *CDY1b* copies present on the Y chromosome with the *b2/b3* deletion plus *b2/b4* duplications that represented an oligozoospermia case. However, the effect of this conservative substitution is unclear.

There was thus no evidence that the sequence variation in *DAZ*, *BPY2*, and *CDY* genes has any effect on infertility related parameters in the subjects examined.

## Discussion

We conducted a comprehensive investigation of partial deletion subtypes of the Y-chromosomal *AZFc* region in 2324 Estonian men, approximately half with idiopathic spermatogenic impairment (n = 1190) in comparison to the reference group (n = 1134). Importantly, 2000 men had undergone full and uniformly conducted andrological workup at a single clinical center, facilitating fine-scale genotype–phenotype analysis. Previously, no study had undertaken re-sequencing of the retained *DAZ*, *BPY2*, and *CDY* genes along with the assessment of the Y haplogroup, dosage, and retained/deleted genes in the *gr/gr* or *b2/b3*-deleted chromosomes in both infertile men and controls. Concordant with the reports from other European populations, the *gr/gr*, but not the *b2/b3* deletion, is a risk factor for spermatogenic impairment in Estonian men with >2-fold increased susceptibility to infertility (*Figure 1D*). However, the gathered data on the large group of reference men in the current study demonstrated the existence of Y chromosomes carrying a *gr/gr* deletion without any documented effect on andrological parameters (*Supplementary file 3*). As a novel finding, the study uncovered complex *AZFc* rearrangements within a specific Y haplogroup, R1a1-M458 and its sub-lineages, causing severe spermatogenic failure in the majority of carriers (*Figure 3*, *Table 3*). This Y lineage has undergone a ~1.6 Mb *r2/r3* inversion in the *AZFc* region that has disrupted the structure of the palindromes P1 and P2, promoting subsequent recurrent deletions and consequently, severely impaired the process of spermatogenesis.

Consistent with key early observations (*Rozen et al., 2012*; *Krausz et al., 2009*; *Machev, 2004*), this study supports the recurrent nature and high subtype diversity of the *AZFc* partial losses that

are currently considered jointly under the umbrella term 'gr/gr deletions'. The 44 detected gr/gr deletions in our study sample were estimated to have originated independently at least 26 times across the Y phylogenetic tree and include seven different combinations of *DAZ* and *CDY1* gene losses. Apparently, there is a substantial undescribed heterogeneity in the spread and structure of gr/gr deletions that in turn contributes to the phenotypic variability of the genetic effects. Unexpectedly, one in four Estonian gr/gr deletion carriers belonged to the Y-chromosomal haplogroup R1a1-M458 (and its sub-lineages) (*Figure 2A,B*; 22.7% vs 5.1% reported as the Estonian population frequency; *Underhill et al., 2015*). Notably, a previous study has reported a significant enrichment of the haplogroup R1a (ancestral lineage to the R1a1-M458) among gr/gr-deleted chromosomes in the Polish population (*Rozen et al., 2012*), which has a high prevalence, 25%, of R1a1-M458 (*Underhill et al., 2015*; *Figure 4A*, *Supplementary file 15*). All the Estonian gr/gr cases and also additional b2/b3 deletion chromosomes representing this Y lineage carried unusual retained *DAZ* gene pairs (*DAZ1-DAZ3* or *DAZ2-DAZ4*) in combination with either *CDY1a* or *CDY1b* gene copy (*Figure 2*). These complex *AZFc* rearrangements were best explained by a preceding (and apparently fixed in R1a1-M458) ~1.6 Mb inversion between the homologous r2 and r3 amplicons, followed by recurrent secondary partial *AZFc* deletions (*Figure 3A*). The latter are facilitated by large ampliconic segments positioned in the same orientation. Inversions in the *AZFc* region are not uncommon, but none of the previously described inversions is expected to substantially disrupt the core palindromic structure of the *AZFc* region (*Figure 1—figure supplement 1*; *Repping et al., 2004*; *Machev, 2004*). In contrast, the r2/r3 inversion disrupts the structure of palindrome P1 and expands the size of the P2 palindrome more than twofold (*Figure 3A*). The critical role of intact P1–P2 palindromes in the *AZFc* structure is supported by the observation that no Y chromosomes have been described with a single *DAZ* gene copy, whereas the inverted *DAZ* gene pairs form the 'heart' of both P1 and P2. In future studies, long-read sequencing technologies should be applied to determine the detailed genomic structure of the *AZFc* region in the R1a1-M458 chromosomes and the exact chromosomal breakpoints of the identified r2/r3 inversion and secondary deletion events in oligozoospermia patients.

As a likely scenario, the r2/r3 inversion plus deletion may predispose to spermatogenic impairment through substantial destabilization of the intra-chromosomal structure affecting meiotic

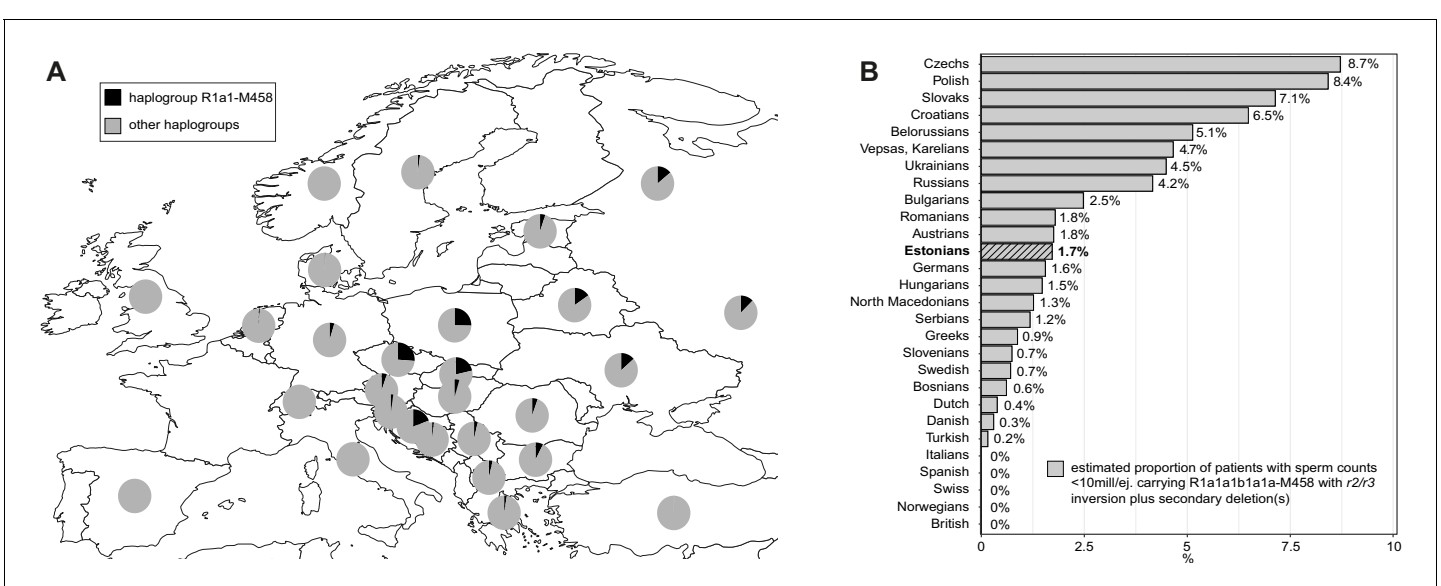

**Figure 4.** The prevalence of the Y-chromosomal haplogroup R1a1-M458 carrying a fixed r2/r3 inversion. (A) Geographical distribution of haplogroup R1a1-M458 and its sub-lineages in Europe. Pie charts indicate populations, with the black sector showing the proportion of R1a1-M458 according to *Underhill et al., 2015*. (B) The estimated proportion of subjects among idiopathic cases with severe spermatogenic failure (sperm counts 0–10 × 10⁶/ejaculate) carrying R1a1-M458 Y lineage (and its sub-lineages) chromosomes that have undergone a subsequent partial *AZFc* deletion. The prevalence was estimated using reported population frequencies of R1a1-M458, including Estonians (*Underhill et al., 2015*) and data available in the current study for Estonian men with spermatogenic failure. Estonians are shown in bold and with a striped filling. For full details, see *Supplementary file 15*.

recombination and chromosomal segregation. The removal of some specific genetic factor(s) being responsible for the phenotypic outcome seems a less likely explanation as the *gr/gr* deletion locations are variable, and so far, no reproducible associations with the exact deleted regions or specific gene copies have been identified. However, recent reports have uncovered an abundance of Y-chromosomal non-coding RNAs and their potential functional involvement in spermatogenesis (*Johansson et al., 2019*). The *AZFc* region contains at least one multicopy family of non-coding RNA genes, *TTTY4* (Testis-Specific Transcript, Y-Linked 4) with testis-enriched expression. These genes are located in the three *g1–g3* duplicons flanking the P1 and P2 palindromic 'hearts'. The phenotypic consequences of *TTTY4* copy number changes are still to be studied. Among 12 Estonian subjects carrying the *AZFc r2/r3* inversion plus deletions and with available data for sperm counts, nine cases exhibited severe spermatogenic failure, two cases had moderate oligozoospermia, and only one case (aged 18 years) was normozoospermic (*Figure 3B,C*). This represented ~8- to 9-fold enrichment of this complex rearrangement among men with severely reduced sperm counts ($0–10 \times 10^6$; *Table 3*). This genetic effect was observed specifically on the effectiveness of spermatogenesis, whereas the measurements of bitesticular volume and reproductive hormone levels did not stand out among the rest of analyzed infertile men. Unfortunately, none of the patients carrying the *r2/r3* inversion plus deletion had undergone a testicular biopsy during their infertility workup. The histopathological pattern of germ cell abnormalities among these cases remains to be investigated in follow-up studies. This knowledge would facilitate understanding of the consequences this Y-chromosomal rearrangement on spermatogenesis and so maximize the benefit of molecular diagnostics in evidence-based clinical management decisions.

To our knowledge, no other Y-lineage-specific risk variants for spermatogenic impairment have been reported so far. Previously, the *DAZ2–DAZ4* deletion had been shown as a high-risk factor for male infertility in the Tunisian population, but the Y haplogroups of those subjects were not investigated (*Ghorbel et al., 2016*). The survival of such a high-risk lineage in the population seems at first sight surprising, but may be accounted for by its possible age-specific effects on spermatogenesis, which may be exacerbated by the recent general decline in sperm count (*Andersson et al., 2008*). In the past, this lineage may not have been disadvantageous. The possible age-related progressive worsening of the reproductive phenotype among *r2/r3* inversion plus deletion carriers should be investigated in follow-up, ideally longitudinal, studies of sufficiently large numbers of patients to make robust conclusions.

This study outcome has notable clinical implications for the improvement of molecular diagnostics and reducing the proportion of idiopathic male factor infertility cases. In Northern and Central Europe, the prevalence of R1a1-M458 haplogroup carrying the *r2/r3* inversion ranges from ~1% in the Netherlands and Denmark to ~2–5% in Austria, Hungary, Germany, Baltics, and most Balkan countries, whereas it is widespread in Slavic populations and carried by 12–26% of men (*Figure 4A*, *Supplementary file 15*; *Underhill et al., 2015*). In non-European populations, the R1a1-M458 Y chromosomes are virtually non-existent (for details, see *Underhill et al., 2015*). However, in some European populations, recurrent secondary *AZFc* partial deletions on Y chromosomes representing the R1a1-M458 haplogroup (and its sub-lineages) may potentially explain from 0.3% up to ~9% of cases presenting severe spermatogenic impairment (sperm counts < 10 million per ejaculate) (*Figure 4B*). Further studies in other populations and large samples of patients and normozoospermic controls are required to fully establish the value of extending the current recommended testing of Y-chromosomal deletions by including the analysis of this novel Y-lineage-specific pathogenic *AZFc* rearrangement.

The evidence from the literature has shown that the increased prevalence of either *gr/gr* or *b2/b3* deletions in infertility cases appears to be population-dependent (*Bansal et al., 2016a*; *Bansal et al., 2016b*). It can be speculated that also in other populations some specific Y lineages may carry *AZFc* structural variants that in combination with partial deletions (or other rearrangements) predispose to chromosomal instability in the complex process of spermatogenesis involving multiple well-coordinated cell divisions. So far, the largest study of the Y-chromosomal phylogeny of *gr/gr* deletion carriers included 152 infertile subjects representing seven countries with different population genetic structures (*Krausz et al., 2009*). However, the number of cases per population was low and the study included only 17 fertile men. Also, the study did not include fine-scale analysis of Y sub-lineages and the retained gene content. Long-range re-sequencing of the whole *AZFc* region in large numbers of men would be the preferred approach to uncover its structural

complexity. Additional pathogenic *AZFc* rearrangements may also exist among Estonian infertile men. Even after omitting the cases carrying a *gr/gr* deletion at the *r2/r3* inversion background, a non-significant enrichment of the remaining *gr/gr* deletion chromosomes can be observed in patients compared to reference men (1.9 vs 1.1%, p<0.1; *Table 2*).

In addition to the main finding, our deep re-sequencing dataset revealed that neither the dosage, sequence variation, nor exact copy of the retained *DAZ*, *BPY2*, and *CDY1* gene showed any detectable effect on spermatogenic parameters. All chromosomes with *AZFc* partial deletions exhibit extremely low overall sequence variation of the retained *DAZ*, *BPY2*, and *CDY* genes. This observation is consistent with previous reports showing low levels of genetic diversity of the human Y chromosome (*Jobling and Tyler-Smith, 2017*) and suggesting that novel variants may be rapidly removed by active gene conversion among Y-chromosomal duplicate genes or selective constraint (*Hallast et al., 2013*; *Rozen et al., 2003*; *Trombetta and Cruciani, 2017*). Among the re-sequenced 382 chromosomes with *b2/b3* deletions, no pathogenic mutations were detected in the single retained *BPY2* and *CDY1* gene copies. At the same time, the high rates of large structural rearrangements and copy number variation in the Y chromosome are well established, contrasting with low levels of sequence variation (*Teitz et al., 2018*; *Shi et al., 2019*). One in five or six Estonian Y chromosomes with *gr/gr* and *b2/b3* deletions had undergone secondary rearrangements with no apparent effect on tested andrological parameters and fertility potential (*Table 2*). In the literature, the data about the effects of secondary duplications after an initial *AZFc* partial deletion on sperm parameters are inconclusive. Some studies have suggested increased pathogenicity (*Lu et al., 2011*; *Yang et al., 2010*; *Lin et al., 2007*; *Ye et al., 2013*; *Yang et al., 2015*), whereas others have reported neutral or even positive effects on spermatogenesis (*Krausz et al., 2009*; *Giachini et al., 2008*; *Lo Giacco et al., 2014*; *Noordam et al., 2011*). However, further copy number reductions in this genomic region appear to be very rare – none of the 44 *gr/gr* or 631 *b2/b3* deletion carriers were identified with further reductions beyond what is expected from the initial deletion.

The current study is the largest and most detailed to date in terms of both the number of patients with spermatogenic impairment and reference samples with available andrological data from a single population, and detailed characterization of the genetic diversity of the *AZFc* region and phylogenetic background of the Y chromosomes. Yet, the total number of identified cases with the *r2/r3* inversion followed by a deletion was relatively small and also inadequate to reach the statistical power in association testing with andrological parameters. Follow-up replication studies utilizing sample cohorts from populations with high(er) R1a1-M458 frequency (e.g. Polish, Czech) should be undertaken to confirm the prevalence and significance of the identified risk variant. The biggest challenge of such studies is the availability of sufficiently large sample collections of both patients and reference cases with andrological data. The identification of men with R1a1-M458 Y chromosomes and characterization of subsequent deletion subtypes only require standard inexpensive laboratory techniques such as PCR, restriction fragment length polymorphism (RFLP) analysis, and Sanger sequencing.

In summary, we have undertaken a comprehensive study of the carriers of *AZFc* partial *gr/gr* and *b2/b3* deletions and uncovered high levels of structural variation in the *AZFc* locus, but low sequence diversity of the coding genes within the region. As a major finding, we discovered a large inversion specific to the Y lineage R1a1-M458 that represents a hotspot for subsequent *AZFc* partial deletions. Men carrying Y chromosomes with this complex rearrangement have >10 fold increased risk of severe spermatogenic failure, but the consequences of this risk could potentially be alleviated by early identification of the variant carriers and facilitating the storage of their sperm samples. Our study results thus have the potential to improve clinical diagnostics and management of idiopathic impaired spermatogenesis in a significant fraction of men originating from Northern and Central European populations.

## Materials and methods

### Study subjects

Patients with idiopathic spermatogenic impairment (n = 1190) were recruited at the Andrology Centre at Tartu University Hospital (AC-TUH) in 2003–2015 (PI: M. Punab). Included cases showed reduced sperm counts ($<39 \times 10^6$/ejaculate) in at least two consecutive semen analyses

(*World Health Organization, 2010*). Recruitment and sampling, semen analyses, hormone assays, and definition of idiopathic cases have previously been described in detail (*Punab et al., 2017*). Men with known causes of male infertility detected during routine diagnostic workup were excluded, for example cryptorchidism, testicular cancer, orchitis/epididymitis, mumps orchitis, testis trauma, karyotype abnormalities, and complete Y-chromosomal microdeletions. The final idiopathic infertility group included 104 azoospermia (no sperm), 88 cryptozoospermia (sperm counts > 0–1 $\times$ 10$^6$/ejaculate), 319 severe oligozoospermia (1–10 $\times$ 10$^6$), and 679 moderate oligozoospermia (10–38 $\times$ 10$^6$) cases (*Table 1*, *Supplementary file 1*).

The reference sample of Estonian men (n = 1134) comprised healthy young men (n = 499) and subjects with proven fatherhood (n = 635) (*Table 1*, *Supplementary file 1*). The cohort of 'Estonian young men' (n = 499) was recruited at the AC-TUH in 2003–2004 (PI: M. Punab), representing a healthy male group with median age 18.6 (17.2–22.9) years at the time of recruitment (*Grigorova et al., 2008*). The subgroup of 'Partners of pregnant women' (n = 324) includes male partners of pregnant women, recruited in 2010–2014 at the Tartu University Hospital and the West Tallinn Central Hospital (*Punab et al., 2017*). Eight hundered and ten men (of 823) in these subgroups underwent sperm analysis.

The subgroup of 'REPROMETA proven fathers' (n = 311) was recruited in 2006–2011 at the Women's Clinic at Tartu University Hospital during the REPROMETA study (PI: M. Laan), originally designed to collect mother–father–placenta trios at delivery to investigate genetics of pregnancy complications (*Kikas et al., 2020*; *Pilvar et al., 2019*). In this study, the REPROMETA fathers represented reference men with proven fertility. Only self-reported age and body mass index (BMI) data were available for this subgroup.

All men, who had turned to Andrology Centre, Tartu University Hospital (AC-TUH) due to idiopathic infertility (n = 1190), as well as the participants of the 'Estonian young men' cohort (n = 499) and the subgroup 'Partners of pregnant women' (n = 324) were offered complete routine andrological workup. The subjects were examined by specialist andrologists at the AC-TUH, who had received respective training in clinical assessment and standardized andrological workup, locally and in collaboration with other European Andrology Academy (EAA)-accredited centers. Also, anthropometric parameters were documented during clinical examination. Details are described in *Punab et al., 2017*.

Physical examination for the assessment of genital pathology and testicular size (orchidometer; made of birch wood, Pharmacia and Upjohn, Denmark) was performed with the patients in standing position. The total testis volume is the sum of right and left testicles. The position of the testicles in the scrotum, pathologies of the genital ducts (epididymitis and ductus deference), and the penis, urethra, presence, and, if applicable, grade of varicocele were registered for each subject.

For 2000 study subjects, sperm analysis was performed, whereas 13 reference cases did not agree with this procedure. Semen samples were obtained by patient masturbation, and semen analysis was performed in accordance with the World Health Organization (WHO) recommendations. In brief, after ejaculation, the semen was incubated at 37°C for 30–40 min for liquefaction. Semen volume was estimated by weighing the collection tube with the semen sample and subsequently subtracting the predetermined weight of the empty tube assuming 1 g = 1 ml. For assessment of the spermatozoa concentration, the samples were diluted in a solution of 0.6 mol/l NaHCO3% and 0.4% (v/v) formaldehyde in distilled water. The spermatozoa concentration was assessed using the improved Neubauer haemocytometers.

Genomic DNA was extracted from EDTA-blood. After blood draw in the morning, serum and plasma fractions were separated immediately for hormone measurements (FSH, LH, testosterone). All laboratory analyses and routine genetic testing (karyotyping, Y-chromosomal microdeletions) were performed at the United Laboratories of Tartu University Hospital according to the established clinical laboratory guidelines. Detailed methodology and reference values for hormonal levels are available by the service provider: https://www.kliinikum.ee/yhendlabor/analueueside-taehestikuline-register.

## Genotyping Y-chromosomal microdeletions

All study subjects (n = 2324) were typed for complete *AZFa* (loss of markers sY84 and sY86), *AZFb* (sY127 and sY134), *AZFc* (sY254 and sY255), and partial *AZFc* deletions *gr/gr* (sY1291), *b2/b3* (sY1191), and *b1/b3* (sY1161, sY1191, and sY1291) following established PCR primers (*Krausz et al.,*

*2014*; *Lin et al., 2006*; *Supplementary file 16*). The multiplex PCR contained the final concentrations of 1× PCR buffer B1 (Solis Biodyne, Estonia), 2.5 mM MgCl$_2$, 2.5 mM dNTP, 2 µM PCR primers for STS markers sY1291 and sY1201, 3 µM PCR primers for STS markers sY1191, sY1206, and sY1161 (*Supplementary file 16*), 1U FIREPol DNA polymerase (Solis Biodyne), and 10 ng of template genomic DNA per reaction. The following PCR conditions were used: for 5 min at 95℃, followed by 32 cycles of 30 s at 95℃, 30 s at 63℃ and 1 min at 72℃, final extension of 10 min at 72℃ and a 4℃ hold. The presence/absence of PCR products in a reaction were checked on 2% agarose gel. Lack of amplification of STS marker sY1291 (but presence of all others) was used to determine the *gr/gr* deletion and lack of sY1191 the *b2/b3* deletion.

## Re-sequencing of retained *DAZ*, *BPY2*, and *CDY* genes using Illumina MiSeq

Re-sequencing of the exonic regions of the retained *AZFc* genes (according to Ensembl release 84, RRID:SCR_002344) in 476 cases with either *gr/gr* or *b2/b3* deletions targeted in total 94,188 bp per subject. *CDY*, *BPY2*, and *DAZ* genes were amplified using 8, 10, or 26 PCR primer pairs, respectively (*Supplementary files 17* and *18*). The presence of all amplicons was confirmed using gel electrophoresis. Amplicons were pooled in equimolar concentrations, barcoded per sample, and sequenced (250 bp reads, paired-end) on Illumina MiSeq (RRID:SCR_016379) with at least 40× coverage. BWA (v0.7.15, RRID:SCR_010910) (*Li and Durbin, 2009*) was implemented to map the sequencing reads to a modified human genome reference (GRChg38), where *CDY1a*, *CDY2a*, *BPY2a* and either *DAZ3-DAZ4* (*gr/gr* carriers) or *DAZ1-DAZ2* (*b2/b3* carriers) remained unchanged, but the sequences of other *CDY*, *BPY2*, and *DAZ* gene copies were replaced with 'Ns'. SNVs and indels were identified using GATK HaplotypeCaller (v3.7, RRID:SCR_001876), with a minimum base quality 20 and outputting all sites (*McKenna et al., 2010*). Y-chromosomal phylogenetic markers were called using bcftools (v1.8, RRID:SCR_005227), with minimum base quality 20, mapping quality 20 and defining ploidy as 1.

Re-sequencing included 31 patients and nine reference men with *AZFc gr/gr* deletions. Six men carrying *gr/gr* deletions were not analyzed due to DNA limitations (two cases) or unavailable andrological data (four cases). The analysis of *b2/b3* deletion carriers included 382 patients (haplogroup N3: n = 380; non-N3, n = 2) and 54 'Partners of pregnant women' (N3: n = 53; non-N3, n = 1).

## Analysis of variant effects from the illumina MiSeq dataset

Variant effect prediction was performed using the Variant Effect Predictor tool (VEP, https://www.ensembl.org/Tools/VEP, Ensembl release 99, RRID:SCR_007931) (*McLaren et al., 2016*). The Combined Annotation Dependent Depletion (CADD) score ≥ 20, that is, including variants among the top 1% of deleterious variants in the human genome, was considered indicative of potential functional importance of identified SNVs in the coding regions (*Rentzsch et al., 2019*).

## Y-chromosomal haplogroup typing

Y lineages of the *gr/gr* samples were defined using 14 markers included in the re-sequencing, plus 34 additional markers determined by Sanger sequencing or restriction fragment length polymorphism (RFLP) analysis (*Supplementary files 18* and *19*). The *b2/b3*-deletion carriers were typed for Y marker N3-M46 (Tat) (*Zerjal et al., 1997*). The sub-lineages of the re-sequenced haplogroup N3 samples were defined in more detail using 16 phylogenetic markers from the Illumina MiSeq dataset, following established nomenclature (*Ilumäe et al., 2016*; *Karmin et al., 2015*). For the other haplogroups, nomenclature according to the International Society of Genetic Genealogy (ISOGG, version 14.14) was followed.

The R1a1a1b1a1a-lineage-specific phylogenetic marker M458 (rs375323198, A > G polymorphism, GRCh38 genomic coordinate: chrY: 22220317), indicating the carrier status of *r2/r3* inverted Y chromosome was amplified using the following conditions: PCR contained the final concentrations of 1× PCR buffer B1 (Solis Biodyne), 2.5 mM MgCl$_2$, 2.5 mM dNTP, 10 µM forward and reverse PCR primers for M458 (see *Supplementary file 19* for primer sequences), 1U FIREPol DNA polymerase (Solis Biodyne OÜ), and 10 ng of template genomic DNA per reaction. The following PCR conditions were used: for 5 min at 95℃, followed by 32 cycles of 30 s at 95℃, 30 s at 52℃ and 1 min at 72℃, final extension of 10 min at 72℃ and a 4℃ hold. The presence of the M458 marker in derived state

(instead of the ancestral allele 'A' presence of allele 'G' at position 147) was determined using Sanger sequencing.

## Determination of *DAZ*, *BPY2*, and *CDY* gene dosage and gene types

The Bio-Rad QX 200 Droplet Digital PCR system (RRID:SCR_019707) was used to quantify the copy numbers of the retained *DAZ*, *BPY2*, and *CDY* genes for 44 *gr/gr* deletion carriers (31 cases, 13 reference men with sperm analysis data) and 631 *b2/b3* carriers (382 cases/249 reference men) (*Figure 2—figure supplement 1*). The PCR primers and probes were designed using Primer3plus (version 2.4.2, RRID:SCR_003081), PCR were performed according to the recommendations in the Droplet Digital PCR Application Guide (Bio-Rad, U.S.) (*Supplementary file 20*) and described in *Shi et al., 2019*. A XQ200 Droplet Reader was used to measure the fluorescence of each droplet and QuantaSoft software (v1.6.6.0320; Bio-Rad) to cluster droplets into distinct fluorescent groups. The copy number of each gene was determined by calculating the ratio of target (unknown – *DAZ*, *BPY2*, or *CDY*) and reference (single-copy *SRY* gene) concentration. ddPCR for each gene were performed once for every sample. For samples carrying the *gr/gr* deletion, if the copy number obtained differed from the expected (two copies of *DAZ* and *BPY2*, three copies of *CDY*), then the ddPCR reaction was repeated. For *b2/b3* carriers, typing was repeated for all samples not carrying the two most typical copy numbers (2-1-3 or 4-2-4 copies of *DAZ*, *BPY2*, and *CDY* genes, respectively). Additionally, a total of 5% of random samples were replicated. If the copy number estimates between replicates differed by 0.8 or more, then a third replicate was performed, and the final copy number was calculated as average of the two closest replicates.

The re-sequencing data of the *DAZ* genes covered nine paralogous sequence variants that were used to determine the retained gene copies in the *gr/gr* and *b2/b3* deletion carriers (*Supplementary file 21*). For the validation of the *DAZ* gene copy mapping approach, at least five *gr/gr* carriers were additionally typed for published SNV combinations differentiating the *DAZ* gene copies (*Machev, 2004*; *Fernandes et al., 2002*). The retained *CDY1* gene was identified according to *Machev, 2004*.

## Genetic association testing with andrological parameters

Statistical testing for the associations between *AZFc gr/gr* or *b2/b3* deletions and andrological parameters was conducted using RStudio (version 1.2.1335, RRID:SCR_000432), and data were visualised using ggplot2 (version 3.2.1, RRID:SCR_014601) (*Wickham, 2009*). Differences in continuous clinical variables between groups were compared using the non-parametric pairwise Wilcoxon rank-sum test.

Genetic association with the carrier status of *b2/b3* deletion and its subtypes was also tested using linear regression analyses adjusted for age. For sperm parameters abstinence time and for total testosterone levels, BMI estimates were additionally used as cofactors. Natural log transformation was used to achieve an approximately normal distribution of values. In all cases (except total sperm counts), the applied transformation resulted in a close-to-normal distribution of values. For the linear regression analyses, statistical significance threshold after correction for multiple testing was estimated $p < 1.0 \times 10^{-3}$ (six tests × eight independent parameters).

## Acknowledgements

We thank all the patients for making this study possible. The clinical team at the Andrology Centre, Tartu University Hospital is thanked for the professional phenotyping and assistance in patient recruitment over many years. Mart Adler and Eve Laasik are specifically acknowledged for the management of the Androgenetics Biobank, and all ML team members are thanked for their contributions to the DNA extractions. Our work was supported by the Wellcome Trust (098051). This work was supported by Estonian Research Council Grants PUT1036 (PH and LK), PUT181 (MP, KP, and OP), IUT34-12 (LK, MG, KR, and ML), PRG1021 (MP, KR, and ML), IUT24-1 (RF and SR), and MOBTT53 (SR). Establishment of the cohort of Estonian infertility subjects was also supported by the EU through the European Regional Development Fund, project HAPPY PREGNANCY, no. 3.2.0701.12–004 (ML, MP, and KR).

## Additional information

### Funding

| Funder | Grant reference number | Author |
| --- | --- | --- |
| Estonian Research Council | PUT1036 | Pille Hallast<br>Laura Kibena |
| Estonian Research Council | PUT181 | Margus Punab<br>Olev Poolamets<br>Kristjan Pomm |
| Wellcome Trust | 098051 | Pille Hallast<br>Laura Kibena<br>Elena Arciero<br>Yali Xue<br>Chris Tyler-Smith |
| Estonian Research Council | IUT34-12 | Laura Kibena<br>Marina Grigorova<br>Kristiina Rull<br>Maris Laan |
| Estonian Research Council | IUT24-1 | Siiri Rootsi<br>Rodrigo Flores |
| Estonian Research Council | MOBTT53 | Siiri Rootsi |
| European Regional Development Fund | 3.2.0701.12-004 | Margus Punab<br>Kristiina Rull<br>Maris Laan |
| Estonian Research Council | PRG1021 | Margus Punab<br>Kristiina Rull<br>Maris Laan |

The funders had no role in study design, data collection and interpretation, or the decision to submit the work for publication.

### Author contributions

Pille Hallast, Conceptualization, Resources, Data curation, Formal analysis, Supervision, Funding acquisition, Validation, Investigation, Visualization, Methodology, Writing - original draft, Project administration, Writing - review and editing; Laura Kibena, Data curation, Formal analysis, Validation, Investigation, Methodology, Writing - review and editing; Margus Punab, Resources, Data curation, Supervision, Writing - review and editing; Elena Arciero, Investigation, Writing - review and editing; Siiri Rootsi, Marina Grigorova, Rodrigo Flores, Olev Poolamets, Kristjan Pomm, Paul Korrovits, Kristiina Rull, Resources, Data curation, Writing - review and editing; Mark A Jobling, Resources, Writing - review and editing; Yali Xue, Resources, Supervision, Writing - review and editing; Chris Tyler-Smith, Resources, Supervision, Funding acquisition, Writing - review and editing; Maris Laan, Resources, Formal analysis, Supervision, Funding acquisition, Investigation, Writing - original draft, Writing - review and editing

### Author ORCIDs

Pille Hallast (iD) https://orcid.org/0000-0002-0588-3987
Maris Laan (iD) https://orcid.org/0000-0002-8519-243X

### Ethics

Human subjects: The study was approved by the Ethics Review Committee on Human Research of the University of Tartu, Estonia (permissions 146/18, 152/4, 221/T-6, 221/M-5, 272/M-13, 267M-13, 286M-18, 288M-13), and sequencing/genotyping was approved at the Wellcome Sanger Institute under WTSI HMDMC 17/105. Written informed consent for evaluation and use of their clinical data for scientific purposes was obtained from each person prior to recruitment. All procedures and methods have been carried out in compliance with the guidelines of the Declaration of Helsinki.

Decision letter and Author response
Decision letter https://doi.org/10.7554/eLife.65420.sa1
Author response https://doi.org/10.7554/eLife.65420.sa2

## Additional files

### Supplementary files

• Supplementary file 1. Characteristics of the Estonian patients with idiopathic spermatogenic impairment and the used reference groups showing mean and standard deviation values.

• Supplementary file 2. Frequencies of *AZFc* partial deletions identified in patients and reference groups.

• Supplementary file 3. Andrological parameters of patients and reference cases with and without the *AZFc* rearrangements.

• Supplementary file 4. Genetic association test with the carrier status of *AZFc b2/b3* deletion results using linear regression.

• Supplementary file 5. Y haplogroup distribution and enrichment of lineage R1a1a1b1a1a-M458 among Estonian patients and reference cases carrying the gr/gr deletions. (a) Y haplogroup distribution of the Estonian patients with idiopathic spermatogenic impairment and reference cases carrying *gr/gr* deletions.(b) Enrichment of Y-chromosomal lineage R1a1a1b1a1a-M458 in men carrying *gr/gr* deletion.

• Supplementary file 6. Y haplogroup distribution of the Estonian patients with idiopathic spermatogenic impairment and reference cases with Y chromosomes having lost the *b2/b3* deletion marker sY1191.

• Supplementary file 7. Retained *DAZ*, *BPY2*, and *CDY1* copy numbers on the Y chromosomes with either *gr/gr* deletion or having lost the *b2/b3* deletion marker sY1191.

• Supplementary file 8. Retained *DAZ*, *BPY2*, and *CDY1* copy numbers on the Y chromosomes with either *gr/gr* deletion or having lost the *b2/b3* deletion marker sY1191.

• Supplementary file 9. Retained *DAZ*, *BPY2*, and *CDY1* copy numbers according to Y lineage in samples having lost the *b2/b3* deletion marker sY1191.

• Supplementary file 10. Deleted *DAZ* and *CDY1* gene types in *gr/gr* and *b2/b3* carriers.

• Supplementary file 11. Detailed copy number, deletion type, and Y haplogroup information for samples carrying the *gr/gr* deletion.

• Supplementary file 12. Andrological parameters of 10 patients and two reference cases with 'r2/r3' inversion plus *gr/gr*, *b2/b3*, or complex deletion.

• Supplementary file 13. Summary of genetic variation identified on the Y chromosomes with *AZFc* region rearrangements (n = 476).

• Supplementary file 14. Identified SNVs and indels from re-sequencing of retained *DAZ*, *BPY2*, and *CDY* genes on the Y chromosomes with *AZFc* region rearrangements.

• Supplementary file 15. Population frequencies of R1a1a1b1a1a-M458 Y lineage and expected proportion of cases with the complex rearrangement *r2/r3* inversion + secondary rearrangement among men with sperm counts of <10 mill/ejaculate.

• Supplementary file 16. Y-chromosomal STS markers and PCR primers used for detection of partial *AZFc* deletions.

• Supplementary file 17. Genomic coordinates of regions sequenced using Illumina MiSeq.

• Supplementary file 18. PCR primers and reaction conditions to amplify regions of interest for sequencing with Illumina MiSeq.

• Supplementary file 19. PCR primers used for typing of Y phylogenetic markers.

• Supplementary file 20. PCR primers and probes used for copy number detection of *DAZ*, *BPY2*, and *CDY* genes using droplet digital PCR.

- Supplementary file 21. Paralogous sequence variants used to determine the retained *DAZ* and *CDY1* genes.
- Transparent reporting form

## Data availability

Illumina MiSeq re-sequencing data are available through the European Genome-phenome Archive (EGA, https://www.ebi.ac.uk/) under the accession number: EGAS00001002157.

The following dataset was generated:

| Author(s) | Year | Dataset title | Dataset URL | Database and Identifier |
|---|---|---|---|---|
| Hallast P, Kibena L, Punab M, Laan M, Xue Y, Tyler-Smith C | 2017 | Resequencing candidate genes for male spermatogenic impairment | https://ega-archive.org/datasets/EGAD00001006784 | EGA, EGA S00001002157 |

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
