## [Decision Letter]

**Acceptance summary:**

This study presents extensive genetic analysis of a relatively large cohort of men with idiopathic infertility, with considerable accompanying andrological phenotypic data. Through careful step-by-step investigations, an inversion variant is identified as a risk factor for subsequent deletion variants that can lead to substantially increased risk of impaired spermatogenesis, on an age-structured basis, relative to non-carriers. As part of the most comprehensive investigation of *AZFc* microdeletions and structural variation to date, the authors have identified a novel structural variant of the Y-chromosome that predisposes to spermatogenic failure and provided clear guidelines for genetic counselling. This work will be of particular interest to the reproductive genetics field, but also has wide ranging implications for colleagues interested in common disease genetics, meiosis, structural variation, dosage sensitivity, and sex chromosome evolution.

**Decision letter after peer review:**

Thank you for submitting your article "A common 1.6 Mb Y-chromosomal inversion predisposes to subsequent deletions and severe spermatogenic failure in humans" for consideration by *eLife*. Your article has been reviewed by three peer reviewers, and the evaluation has been overseen by George Perry as the Senior Editor and Reviewing Editor. The reviewers have opted to remain anonymous.

Essential Revisions:

1) Please compare the estimated cohort allele frequencies for each SV against corresponding gnomAD SV allele frequency estimates (https://gnomad.broadinstitute.org/; https://doi.org/10.1038/s41586-020-2287-8). If the novel risk alleles are represented in gnomAD SV, then this would help alleviate uncertainty around the genotyping. Furthermore, an estimate of the *r2/r3* inversion variant in both European and non-European populations would add important context and potential impact of the inversion+deletion risk allele. If any alleles are not represented in gnomAD SV the authors should comment as to why.

2) In general the primary genotyping data are underrepresented in the figures and supplement. For example, a histogram depicting estimated copy number from the ddPCR experiments would be useful. Good separation of the signal distributions centered around biologically interpretable copy numbers would indicate that the assay is well calibrated with minimal genotyping errors.

3) Are histopathology data from testis biopsies (e.g., see https://doi.org/10.1056/NEJMoa1406192) for cases with the risk allele available? This material may already be banked as part of the infertility workup. If so reporting of these results would be welcomed to help support the model of meiotic failure over gene dosage changes. If not, please mention this as a follow-up opportunity in your Discussion.

4) Please also discuss the potential future benefit (unless these data can be readily generated now) of long-read sequencing (e.g., ONT, PacBio) data to help resolve the relative orientations of the amplicons in one of their inversion+deletion patients. (A traditional cytogenetics approach (i.e., FISH with multiple colored probes) would also work but would require additional biological material). In particular, the long-read sequencing based approach has the added benefit of nominating a putative inversion breakpoint that could be fine-mapped and validated. The authors infer that the *r2/r3* inversion is a single, fixed lineage-specific event, but breakpoint information in multiple individuals would much more rigorously rule out a recurrent rearrangement scenario.

5) Also with respect to the request for slightly expanded Discussion, please offer some comments on potential approaches for (and or challenges associated with) replicating this result in another population/sample.

6) In the Results, please provide more explicit links to the specific assays/methods used to generate them, to aid reader understanding. For example, providing a short phrase that ddPCR (with a half-sentence description of the method) was used to quantify the *DAZ*, *BPY2*, and *CDY* genes. This information is laid out in the Materials and methods but adding to the Results will make the paper more readable. In addition, while the complexity and existing nomenclature of the AZF region contribute to the density of information in the Results section, as you identify descriptive wording that could be trimmed/compressed for readability, please make such changes.

7) Are the Pairwise Wilcoxon Rank Sum Test P-values in Figure 3 and Supplementary file 3 corrected for multiple tests? It's stated that the linear regression for Supplementary file 4 accounts for multiple tests only. Some P-values values in Supplementary file 3 are in bold but unless I'm missing it, it's not stated why.

---

## [Author Response]

Essential Revisions:1) Please compare the estimated cohort allele frequencies for each SV against corresponding gnomAD SV allele frequency estimates (https://gnomad.broadinstitute.org/; https://doi.org/10.1038/s41586-020-2287-8). If the novel risk alleles are represented in gnomAD SV, then this would help alleviate uncertainty around the genotyping.

In gnomAD (v2.1) database, 15,708 whole-genome sequences have been targeted for SV calling. For the Y-chromosome, the number is 3-fold lower as the number of sequenced men is only ~5500. The Y-chromosomal haplogroup composition of these men is not known, but given the very broad demographic spread (~1700 Europeans, ~700 East Asians, ~2600 Africans etc), this is expected to be extremely diverse. Of note, the prevalence of *gr/gr* and *b2/b3* partial *AZFc* deletions is variable across populations and Y-chromosomal lineages, from fixed events on some Y-lineages to no known deletions on other lineages.

A total of 410 Y-chromosomal deletions are included in the gnomAD (v2.1) database, ranging in size from 51 bp to only two longer than 1 Mb (1.07 Mb and 3.6 Mb). Neither of these two >1 Mb-long deletions match the size and/or location of well-established *gr/gr* and *b2/b3* deletions, although both of these are fixed in some Y-lineages. Also, in the current gnomAD SV database, no Y-chromosomal inversions have been mapped.

Based on the affected gene content, two duplication variants in the gnomAD SV database correspond well to complex structural variants included in the current manuscript: the *b2/b4* duplication following the *gr/gr* deletion (ID: DUP_Y_55084; 114 subjects in gnomAD; length 2,105,101 bp, gain of 2 *DAZ*, 2 *BPY2* and 1 *CDY1* genes) and the *b2/b4* duplication following the *b2/b3* deletion (ID: DUP_Y_55085; 58 subjects in gnomAD; length 1,589,100 bp, gain of 2 *DAZ*, 1 *BPY2* and 1 *CDY1* genes). We have added Supplementary file 8 with the frequency data of these variants in different populations and extended the text in the respective Results section.

A possible explanation of the limited representation of the *AZFc* structural variants in gnomAD:

The Y-chromosomal *AZFc* region is composed of six amplicon units with ≥99.94% nucleotide identity, repeated in direct and inverted orientations from two to four times (Figure 1B in the manuscript). The whole region nearly lacks unique genomic sequence and identifying SVs from short-read WGS data is challenging, requiring care and custom approaches that take into account the expected number of each amplicon unit according to human Y chromosome reference sequence (see Teitz et al. 2018). In general, the analytical tools developed for regions with mostly unique sequence do not perform well for the analysis of SVs in highly duplicated genomic regions undergoing constant gene conversion keeping their DNA sequences (nearly) identical.

As an example, when conventional short-read WGS data of the Y-chromosome with two (out of four) deleted *DAZ* gene copies is mapped to the reference sequence, it would appear that all four *DAZ* genes were still present but with slightly reduced read depth. There are very few unique sequence positions that differ between the highly homologous *DAZ* genes (and other segments in the *AZFc* region, see Figure 1B) and the conventional sequence alignment tools lack sensitivity for their detection.

However, with the careful custom approach taken in the current study that included, first, the determination of the gene copy numbers, and then re-sequencing of the retained genes and targeted genotyping of gene-specific paralogous sequence variants, it was possible to determine the exact present and lost gene copies.

Furthermore, an estimate of the r2/r3 inversion variant in both European and non-European populations would add important context and potential impact of the inversion+deletion risk allele. If any alleles are not represented in gnomAD SV the authors should comment as to why.

We have explained above why the *r2/r3* inversion variant is not present in gnomAD SV. In addition, we have shown that *r2/r3* inversion is most probably fixed in one specific European Y-chromosomal lineage. The frequency of this lineage in different European populations is provided in Supplementary file 15 and on Figure 4A, ranging from 0 in many populations to over 20% in some Central and Eastern European populations. The frequency data originate from Underhill et al., 2015, the most comprehensive investigation of the distribution of R1a-M420 sub-lineages to date, using a total of 16,244 male samples from 126 Eurasian populations and showing that R1a1-M458 is virtually absent outside of Europe.

2) In general the primary genotyping data are underrepresented in the figures and supplement. For example, a histogram depicting estimated copy number from the ddPCR experiments would be useful. Good separation of the signal distributions centered around biologically interpretable copy numbers would indicate that the assay is well calibrated with minimal genotyping errors.

We thank the reviewer for the comment and have now included a new Figure 2—figure supplement 1 showing the distribution of raw estimated *DAZ*, *BPY2* and *CDY* copy numbers using droplet digital PCR. Figure 2—figure supplement 1 shows that the raw estimates correspond well to biologically interpretable copy numbers except for some uncertainty among the highest numbers.

3) Are histopathology data from testis biopsies (e.g., see https://doi.org/10.1056/NEJMoa1406192) for cases with the risk allele available? This material may already be banked as part of the infertility workup. If so reporting of these results would be welcomed to help support the model of meiotic failure over gene dosage changes. If not, please mention this as a follow-up opportunity in your Discussion.

In infertility clinics, testis biopsy is mostly used for men with azoospermia. The intention of this invasive procedure is to possibly detect and retrieve immature sperms (through the TESE procedure) that can be used for ICSI to achieve the fertilization of the partner’s oocyte. Among the Estonian patients carrying *r2/r3* inversion plus deletion, there was only one azoospermia patient, but he had not undergone testicular biopsy. Our Discussion has been extended to include this limitation, stressing the importance of testicular histopathological phenotype assessment in future studies to understand the consequences of this rearrangement for spermatogenesis and to maximize the benefit of molecular diagnostics.

4) Please also discuss the potential future benefit (unless these data can be readily generated now) of long-read sequencing (e.g., ONT, PacBio) data to help resolve the relative orientations of the amplicons in one of their inversion+deletion patients. (A traditional cytogenetics approach (i.e., FISH with multiple colored probes) would also work but would require additional biological material). In particular, the long-read sequencing based approach has the added benefit of nominating a putative inversion breakpoint that could be fine-mapped and validated. The authors infer that the r2/r3 inversion is a single, fixed lineage-specific event, but breakpoint information in multiple individuals would much more rigorously rule out a recurrent rearrangement scenario.

We have extended the Discussion by adding such future actions to resolve the detailed structure of R1a1-M458 chromosomes and the breakpoints of *r2/r3* inversion and secondary deletions using long-read sequencing.

In regards to whether the *r2/r3* inversion is fixed or represents a recurrent event in the R1a1-M458 and its sub-lineages, we agree that formally the scenario of recurrent events cannot be rejected. However, with the available phylogenetic information, the possibility of recurrent inversion events is highly unlikely as the Y chromosomes of all the samples with the proposed inversion are closely related and descendants of a single, R1a1-M458 lineage. In support to the lineage-specific fixed event, the *r2/3* inversion was not identified in any other Y-chromosomal lineages in our large sample set.

The low probability of recurrent inversions in R1a1-M458 and its sub-lineages is supported by the split times of these Y chromosomes and estimated mutation rate for Y-chromosomal inversions. It is not possible to estimate the split times of the analysed Y chromosomes and using the data generated in this study; however, we can take advantage of publicly available whole-genome sequenced samples belonging to the same Y-chromosomal lineages (R1a1-M458 and its sub-lineages). The recently published analysis of the whole-genome sequenced Human Genome Diversity Project (HGDP) panel contains 5 males carrying sub-lineages of R1a1-M458 (Bergström et al. 2020, doi:10.1126/science.aay5012), with the time to most recent common ancestor (TMRCA) estimated at approximately 5,000 years.

Unfortunately, no estimated inversion rate is available for the *AZFc* region, but a mutation rate of approximately 2.3 x 10^-4^ events per father-to-son Y transmission has been estimated for recurrent Y-chromosomal IR3/IR3 inversions and can perhaps be used as a rough proxy (Repping et al. 2006; doi:10.1038/ng1754). This rate translates to roughly 1 inversion per 4,347 generations, or using a 30-year male generation time, approx. 1 event per 130,434 years.

Therefore, even if the inversion rate in the *AZFc* region is substantially higher than that estimated for the IR3/IR3 repeats, the occurrence of multiple inversions among Y chromosomes with such a recent TMRCA seems highly unlikely.

5) Also with respect to the request for slightly expanded Discussion, please offer some comments on potential approaches for (and or challenges associated with) replicating this result in another population/sample.

We have extended the Discussion section to suggest replication of the results in other populations, and the possible challenges associated with it.

6) In the Results, please provide more explicit links to the specific assays/methods used to generate them, to aid reader understanding. For example, providing a short phrase that ddPCR (with a half-sentence description of the method) was used to quantify the DAZ, BPY2, and CDY genes. This information is laid out in the Materials and methods but adding to the Results will make the paper more readable.

The Results sections were extended to include such statements about the specific assays and methods used.

In addition, while the complexity and existing nomenclature of the AZF region contribute to the density of information in the Results section, as you identify descriptive wording that could be trimmed/compressed for readability, please make such changes.

We have modified the Results and Discussion sections when talking about the *r2/r3* inversion followed by secondary deletions as “*r2/r3* inversion plus deletion” for smoother readability.

7) Are the Pairwise Wilcoxon Rank Sum Test P-values in Figure 3 and Supplementary file 3 corrected for multiple tests? It's stated that the linear regression for Supplementary file 4 accounts for multiple tests only. Some P-values values in Supplementary file 3 are in bold but unless I'm missing it, it's not stated why.

The Pairwise Wilcoxon Rank Sum Test *p*-values presented in Figure 3 and Supplementary file 3 were not corrected for multiple tests. To make this obvious to the reader, we have now included the following sentence in the figure and table legends, stating the statistical significance threshold after correcting for multiple testing: "Statistical significance threshold after correction for multiple testing was estimated *P*<1.0 x 10^-3^ (a total of 5 tests x 10 independent parameters)". We have also added an explanation to Supplementary file 3 about *p*-values shown in bold. Further, in the Results section “*r2/r3* inversion promotes recurrent…” it is now explicitly said that the reported test result reflects a nominal *p*-value.